Workshop at the 6th Symposium on Advances in Approximate Bayesian Inference (non-archival), 2024 1–7

# Word Embedding Uncertainty Estimation

## 1. Introduction

Word embeddings, which represent words as numerical vectors, can be used to capture statistical relationships between words. While useful, traditional word embedding methods only provide point estimates for each word, thus, there is no direct indication of uncertainty. In many scientific applications, quantifying the uncertainty associated with these estimations is essential for reliable interpretations of the results.

Probabilistic word embeddings extend traditional word embeddings by using priors, and formulate word embeddings as a full probabilistic model. However, the high dimensionality poses difficulties for uncertainty estimation using standard Bayesian inference methods such as Hamiltonian Monte Carlo (Neal et al., 2011). To illustrate, a typical dataset could easily have 100,000 unique words and a 100-dimensional embedding space, resulting in millions of parameters (Mikolov et al., 2013).

There have been methods for embedding uncertainty estimation using bootstrap (Antoniak and Mimno, 2018). However, bootstrap estimation of uncertainty is requires orders of magnitude more computation than obtaining point estimates, which is often not feasible for large datasets. In the Bayesian realm, Bamler and Mandt (2017) propose mean-field variational inference of the embeddings. While being computationally efficient, it is unclear if the variational approximation is reasonable.

## 2. Uncertainty Estimation for Word Embeddings

In this article, we develop uncertainty estimation for probabilistic word embeddings (Rudolph et al., 2016; Bamler and Mandt, 2017; Rudolph and Blei, 2017), which are probabilistic formulations of the skip-gram with negative samples (SGNS) model first presented by Mikolov et al. (2013). Negative samples makes the embeddings more computational efficient, which makes the estimation on large datasets practically feasible (Mikolov et al., 2013).

Let $\mathcal{D}$ denote the dataset, a sequence of words $(w_i)_{i=1}^N$, where $w_i$ is an element of the *vocabulary* $W = \{w_j\}_{j=1}^V$ of all the relevant words of the corpus. For each word $w_j$ in the vocabulary we have a corresponding target $\rho_j \in \mathbb{R}^d$ and context $\alpha_j \in \mathbb{R}^d$ representation, where $d \in \mathbb{N}$ is the *dimensionality* of the embeddings. We write $\rho = (\rho_{w_1}, \rho_{w_2}, \ldots, \rho_{w_V})$, $\alpha = (\alpha_{w_1}, \alpha_{w_2}, \ldots, \alpha_{w_V})$ and $\theta = (\rho, \alpha)$, which yields the matrices $\rho \in \mathbb{R}^{V \times d}$, $\alpha \in \mathbb{R}^{V \times d}$ and $\theta \in \mathbb{R}^{2V \times d}$.

The SGNS likelihood nicely factors into terms, where each term only has a handful of parameters

$$\log p(\mathcal{D} \mid \theta) = \sum_{i=1}^N \left( \underbrace{\sum_{v \in C_i^+} \log \sigma(\rho_{w_i}^T \alpha_v)}_{\text{positive samples}} + \underbrace{\sum_{v \in C_i^-} \log(1 - \sigma(\rho_{w_i}^T \alpha_v))}_{\text{negative samples}} \right) \tag{1}$$

where $C_i^+$ is the context window, or the set of positive samples, for word $w_i$. Specifically, the context window is the set of words within a distance $M$ from the center word, i.e. $C_i^+ = \{w_{i-M}, \dots, w_{i-1}, w_{i+1}, \dots, w_{i+M}\}$. The negative samples $C_i^-$ are drawn randomly from the empirical distribution of the words, sometimes with slight adjustments (Mikolov et al., 2013).

### 2.1. Laplace Approximation

Laplace approximation of the posterior around the MAP estimate, $\hat{\theta}$, is a way to approach uncertainty for posteriors where its difficult to sample directly. In Laplace approximation, we approximate the whole posterior with a large multivariate Gaussian distribution

$$\theta \sim N(\hat{\theta}, \Sigma) \tag{2}$$

where we get the covariance from the inverse of the Hessian of the posterior,

$$\Sigma = \mathbf{H}^{-1}(\hat{\theta}). \tag{3}$$

Unfortunately, we cannot invert the whole matrix in reasonable time due to the large number of parameters. However, in many practical applications we are only interested in the relations between a subset of words. Then, we just need to know the inverse of the subset of the Hessian that covers the interactions between those words.

#### 2.1.1. EFFICIENT LAPLACE APPROXIMATION

In order to obtain results with the Laplace approximation, we need to find individual elements of the inverse of a large matrix. We propose a method that exploits the structure of the posterior. Namely, we utilize the sparsity of the likelihood and the prior, as well as the low-rank structure of the nonzero parts of the Hessian due to the likelihood.

In the likelihood, Equation 1, we observe that the positive sample part only appear when two words co-occur in the data, which is not the case for most word pairs. This means that the contribution from the positive samples in the Hessian are going to be zero. In the 50 million token Wikipedia sample, roughly 0.25% of the words co-occur using a window size of 5. The sparsity is illustrated in Figure 1.

Moreover, the Hessian of the positive samples factors into submatrices consisting of outer products. The following are the diagonal and off-diagonal contributions to the Hessian,

$$\frac{\partial^2}{\partial^2 \rho_u} \log \sigma(\rho_w^T \alpha_v) = -\sum_{k=1}^{N} \mathbb{I}(u = w_k) \sum_{v \in C_k^+} \alpha_v \alpha_v^T \sigma(\rho_{w_k}^T \alpha_v) \sigma(-\rho_{w_k}^T \alpha_v) \tag{4}$$

$$\frac{\partial^2}{\partial \rho_w \partial \alpha_v} \log \sigma(\rho_w^T \alpha_v) = -C(w, v) \, \rho_w \alpha_v^T \sigma(\rho_w^T \alpha_v) \sigma(-\rho_w^T \alpha_v) \tag{5}$$

where $C(w, v)$ is the co-occurrency matrix of word $w$ and $v$. As seen in Figure 1, this is more often than not 0. The off-diagonal submatrices, described in Eq. 5, are always rank 1 or 0, which makes them hundreds of times computationally cheaper than full-rank submatrices.

Exploiting these properties, we can formulate a substantially more efficient algorithm for obtaining samples from the Laplace approximated posterior.

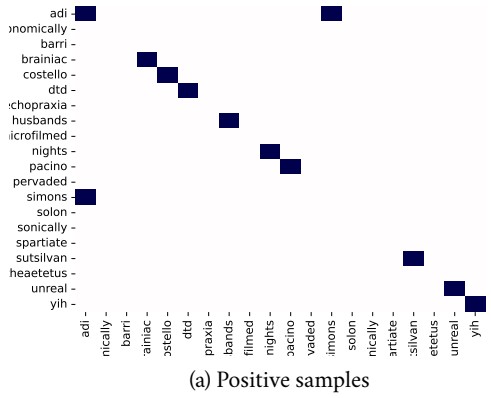

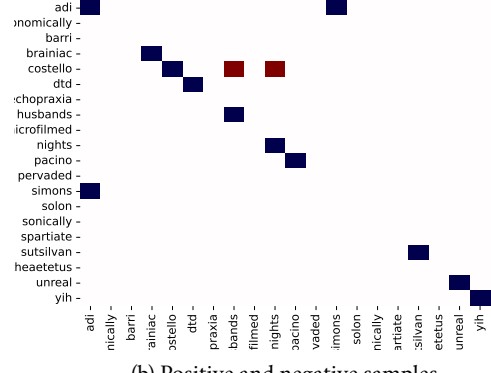

(a) Positive samples (b) Positive and negative samples

Figure 1: Sparsity of the Hessian matrix for a subset of 20 words. Data from a 50M token Wikipedia subset, positive samples in blue. On the right, a random set of negative samples is also included.

## 3. Rotational symmetries of word embeddings

In word embeddings, the likelihood is defined by the dot products. For this reason, the likelihood $\mathcal{L}$ is invariant with respect to orthogonal rotations, i.e.

$$\mathcal{L}(\theta) = \mathcal{L}(O^T \theta) \tag{6}$$

for any orthogonal matrix $O$, $O^T O = I$.

Due to the rotational symmetries, it is not possible to use simple subtraction or Euclidean distance to analyze the difference of two embeddings. This complicates convergence and posterior analysis. The rotational symmetries also make Laplace approximation ill-defined due to a singular Hessian.

### 3.1. Eliminating rotational symmetries

There are

$$\frac{d(d-1)}{2} \tag{7}$$

degrees of freedom in the set of orthogonal matrices $O \in \mathbb{R}^{d \times d}$, $O^T O = I$. Due to this reason, a word embedding of dimensionality $d$ is overparametrized by $\frac{d(d-1)}{2}$ parameters.

For this reason, any set of dot products between the columns of $\theta \in \mathbb{R}^{2V \times d}$ can be achieved by restricting $\frac{d(d-1)}{2}$ of its elements to be zero. Specifically, we formulate $\theta'$ so that the lower diagonal is zero, and other entries are freely variable

$$\theta' = \begin{bmatrix} \theta_{1,1} & \theta_{1,2} & \theta_{1,3} & & \dots & & \theta_{1,2V} \\ 0 & \theta_{2,2} & \theta_{2,3} & & \dots & & \theta_{2,2V} \\ 0 & 0 & \theta_{3,3} & & \dots & & \theta_{3,2V} \\ \dots & \dots & \dots & \dots & & \dots & \\ 0 & \dots & & 0 & \theta_{d,d-1} & \dots & \theta_{d,2V} \end{bmatrix} \tag{8}$$

Theoretically, this correction can be applied to any algorithm. We apply this to VI and comparatively study it with the uncorrected version of the algorithm.

### 3.2. Comparing embeddings with rotational symmetries

Rotation invariant features are commonly used to analyze word embeddings. These include Euclidean distances between the words (column distances) and cosine distances, which are invariant wrt. orthogonal rotations $\theta' = O^T\theta$, where $O^T O = I$. The set of co-occurence probabilities of words $w, v$

$$P(w \wedge v \mid \theta) = \sigma(\alpha_w^T \rho_v) \tag{9}$$

are also invariant to orthogonal rotations. Using that, we define the probability divergence between embeddings $\theta$ and $\theta'$ as the root mean squared error between the co-occurence probabilities

$$d_{co}(\theta, \theta') = \sqrt{\frac{1}{V^2} \sum_{v,w \in W} (P(w \wedge v \mid \theta) - P(w \wedge v \mid \theta'))^2} \tag{10}$$

This divergence is 0 if and only if the embeddings are the same up to a rotation and a uniform scaling. We go on to use this as a similarity metric for two embeddings. This is also the basis of our convergence analysis, if $d_{co}(\hat{\theta}, \theta_{true}) = 0$ we take that the estimate $\hat{\theta}$ has converged to the true parameter $\theta_{true}$.

## 4. Experiments

To compare the estimation methods, we conduct a simulation study with randomly generated word and context vectors. For each $w \in W$, the word and context vectors are simulated from

$$\begin{aligned} \rho_w &\sim \mathcal{N}(0, \varepsilon^2 I/d), \\ \alpha_w &\sim \mathcal{N}(0, \varepsilon^2 I/d), \end{aligned} \tag{11}$$

where $d$ is the dimensionality of the embedding, and the hyperparameter $\varepsilon = 1$. The embedding is scaled inverse to the dimensionality so that the dot products remain roughly similar in magnitude regardless of the dimensionality.

We generate random word pairs $(w, v)$ by sampling uniformly from the set $W$. A Bernoulli random variable $X$ is then sampled

$$\begin{aligned} w &\sim \text{Uniform}(W) \\ v &\sim \text{Uniform}(W) \\ X &\sim \text{Bernoulli}(\sigma(\alpha_v^T \rho_w)) \end{aligned} \tag{12}$$

$N$ times for a simulated dataset with $N$ observations. The simulated data uses up to $N = 100{,}000$ data points, dimensionality of $d = 2$, and a vocabulary of size $V = 10$. This is rather small compared to common practical values of $100 \leq d \leq 300$, $10^6 \leq N \leq 10^9$ and $10^4 \leq V \leq 10^6$.

### 4.1. Experimental Results

As a first step we measure convergence of the methods by looking at the probability divergence $d_{co}$, defined in Equation 10. In Figure 2 we look at how $d_{co}$ evolves as we increase the available number of simulated data points. For both MAP estimation and VI, more data yields results closer and closer to the true embedding $\theta_{true}$, i.e. both $d_{co}(\hat{\theta}_{MAP}, \theta_{true})$ and $d_{co}(\hat{\theta}_{\mu,VI}, \theta_{true})$ decrease. Before 50,000 observations, MAP is better than VI mean, after which the estimators become very similar, as would be expected from Wang and Blei (2018).

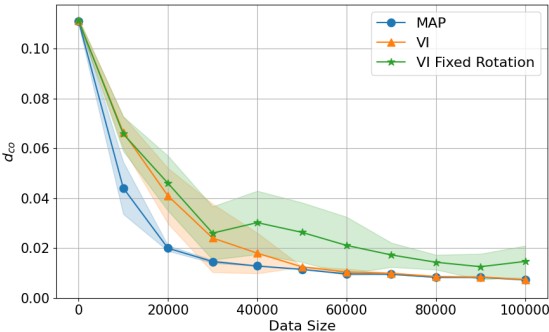

Figure 2: Probability divergence for MAP, VI and VI with rotations eliminated via method described in 3.1, using incrementally larger data to demonstrate convergence. Metric $d_{co}(\hat{\theta}_{MAP}, \theta_{true})$ for the MAP estimate $\hat{\theta}_{MAP}$, and $d_{co}(\hat{\theta}_{\mu,VI}, \theta_{true})$ for the VI mean parameter $\theta_{\mu,VI}$.

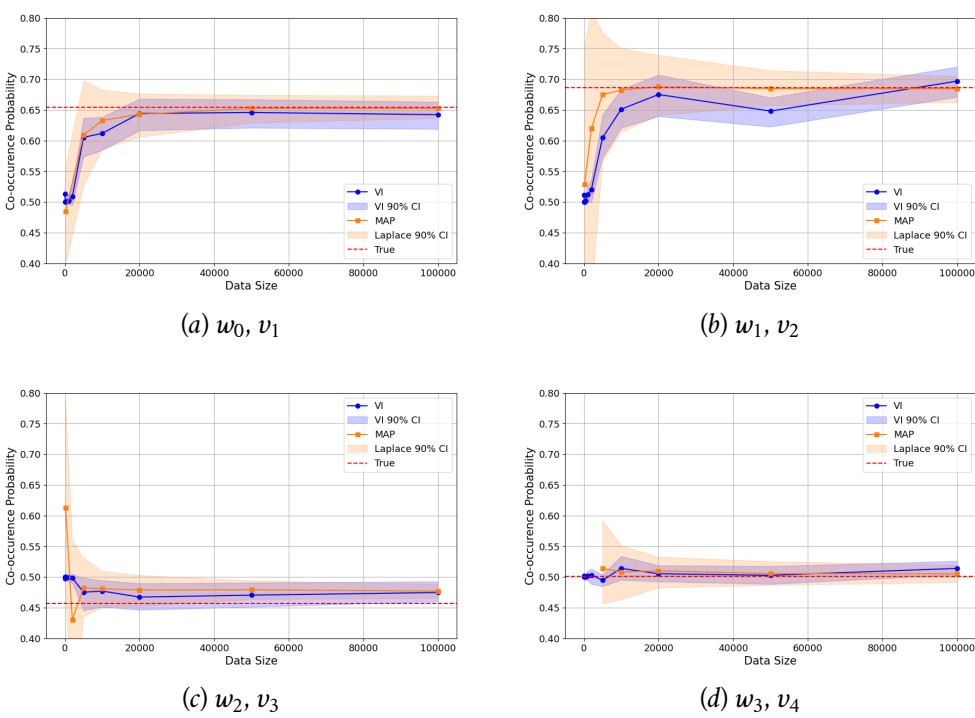

Figure 3: Co-occurence probabilities between two words in our simulated dataset, for 4 pairs of words, using mean field variational inference and Laplace Approximation around MAP, for different data sizes. The blue line represents the estimated mean, the shaded regions around them are the Monte Carlo simulated 90% credible intervals, and finally the red dashed line represents the true co-occurence probability.

In Figure 2, we see that VI with rotational correction yields more inconsistent results. This may be due to the rotation-corrected algorithm being more difficult to optimize, or other numerical problems.

We also study the co-occurence probabilities of specific word pairs. We do this both for a model without any corrections as well as a model with the correction described in 3.1. The results are presented in Figure 3, covering four different word-context pairs.

All of the word similarities for VI seem to tend to the true value as the number of observations increases. However, the CIs for VI do not seem to correspond to the true uncertainty, as they do not get progressively smaller with more data. They also do not seem to consistently contain the true value, especially with small datasets. This is in line with Wang and Blei (2018), who find that the VI mean converges to the true value, while the CIs do not necessarily reflect the true uncertainty.

For MAP/Laplace approximation, the CI bands are wider than with VI, especially with few observations. Moreover, they progressively shrink as the data increases. The credible intervals seem to include the true value more often than for VI. In this sense, the Laplace CIs are better than the VI ones. Some Laplace CIs are missing due to the Hessian not being positive definite, which may be due to optimization issues.

## 5. Conclusion and Future Work

Our early results indicate that Laplace Approximation around MAP provides better estimates in terms of convergence, and more reasonable uncertainty estimates than the MFVI method proposed by Bamler and Mandt (2017).

There are three potential lines of future work. First, we want to run experiments on real data. Secondly, we want to implement a more computationally efficient version of the Laplace approximation, which utilizes the properties of the Hessian that we describe in this article. Finally, we want to compare our results to Hamiltonian Monte Carlo and bootstrap of MAP. While these are not computationally feasible in real life applications, they should give us samples from the true posterior as a benchmark to compare with in a simulation setting.

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
