# OpenReview forum: "Word Embedding Uncertainty Estimation"
_approximateinference.org/AABI/2024/Symposium — AABI 2024_

### Official Review · Reviewer_aHve · 2024-04-13
**Review: Word Embedding Uncertainty Estimation**

**Rating:** 4
**Confidence:** 4

**Review:**

The overall idea of this paper is to develop a new uncertainty estimate of the probabilistic word embeddings from probabilistic SGNS models. This is done by using a laplace approximation of the posterior around the MAP estimate and exploiting the sparsity of the data structure to do this efficiently. This idea seems solid and novel, however, I am not convinced by the numerical results of this paper. The paper is OK written with enough clarity to follow along.

First of all, the paper only presents results on synthetic data which seems very unrealistic compared to practical values. Especially, the number of dimensions simulated (d=2) is concerning and I wonder why they chose this. Thirdly the results in Figs 2 and 3 are not convincing and barely brings anything to the table indicated by the confidence interval, which I appreciate they included. Although the results might indicate slightly better uncertainty estimates than the MFVI, the experimental conditions are not sufficient to make any conclusion about if this statement generalises. The authors should at least argue for why their experiments would generalise to real-world data.

---

### Official Review · Reviewer_TcsE · 2024-04-16
**Uncertainty estimation for word embeddings**

**Rating:** 7
**Confidence:** 2

**Review:**

The manuscript presents a method for estimating uncertainty in word embeddings utilizing the Laplace approximation. The authors compare their proposal to Variational Inference in terms of convergence and uncertainty estimates.

The motivation for the paper could be better justified. Specifically, the authors should elucidate the  necessity of estimating uncertainty in word embeddings, in a way that is not captured by the softmax layer of neural networks.

Additionally, the practical implications of this method could be significantly enhanced by demonstrating its application within a real-world NLP task (the authors acknowledge this in the final section).

In section 4 (“Experiments”), I wasn’t sure about the role of negative sampling in the experiment.

---

### Official Review · Reviewer_8VXh · 2024-04-17
**Word Embedding Uncertainty Estimation**

**Rating:** 5
**Confidence:** 5

**Review:**

Overall, this work explores an interesting and valuable topic: incorporating uncertainty estimation into probabilistic word embedding models. Here's a breakdown of its strengths and weaknesses:

Strengths:

Originality: The paper proposes a novel approach to estimate uncertainty in probabilistic word embeddings using Laplace approximation around the MAP estimate. This addresses a limitation of existing methods that lack efficient uncertainty quantification.
Significance: Uncertainty estimation plays a crucial role in interpreting word embedding results and assessing their reliability. This work can potentially improve the interpretability and trustworthiness of word embeddings used in various NLP tasks.
Technical Soundness: The paper provides a detailed explanation of the proposed method, including its theoretical background and computational considerations. The authors address the challenges associated with high dimensionality and rotational symmetries in word embeddings.
Evaluation Strategy: The work includes a simulation study to compare the proposed method with existing ones.
Weaknesses:

Limited Scope: The evaluation is based on simulated data. It would be more convincing to see results on real-world datasets with larger vocabulary sizes and higher dimensionality.
Computational Efficiency: While the paper discusses the efficiency of the Laplace approximation compared to full Bayesian inference, it would be beneficial to quantify the computational cost compared to existing methods.
Convergence of VI with Rotational Correction: Figure 2 suggests that VI with rotational correction might have convergence issues. This aspect needs further investigation and potential improvement.
Credible Intervals: Although Laplace approximation seems to offer better credible intervals than VI, some intervals are missing due to non-positive definite Hessian. This requires further exploration to ensure reliable uncertainty estimates.
Overall Impression:

This work presents a promising approach to uncertainty estimation in word embeddings. The proposed method has a sound theoretical foundation and demonstrates potential benefits in simulations. However, further evaluation of real-world data and addressing convergence and credible interval issues would strengthen the contribution.

Here are some additional points to consider:

The paper mentions future work directions like implementing a more efficient Laplace approximation and comparing it with other methods like Hamiltonian Monte Carlo. These explorations would be valuable additions to solidify the proposed approach.
The clarity of the writing can be improved in some sections. For example, the notation used for some variables and equations might benefit from clearer explanations for readers unfamiliar with the specific notation.

---

### Official Review · Reviewer_gmBL · 2024-04-24
**Concise clear complete research work**

**Rating:** 7
**Confidence:** 4

**Review:**

In this paper, for efficiently tackling sparsity and uncertainty of word embedding, Laplacian approximation and elimination of rotation symmetry are used. The paper is concise clear complete.

Several suggestions for modification or vision:
1. Because a specific method is selected from the previous work, can the paper change the title with the concrete method?
2. Whether the uncertainty estimation can be implemented in the industrial-level word embedding method will be an interesting question;
3. The propagation of uncertainty of word embedding may have practical meaning for NLP. Perhaps a convincing experiment could be designed.

---

### Meta-Review · Area_Chair_xeLj · 2024-05-26

**Recommendation:** Accept (Poster)
**Confidence:** 4

**Metareview:**

The reviewers have shared detailed feedback and highlighted several strengths and weaknesses of the paper. Three reviewers recommended acceptance and one reviewer recommended rejection. I believe that the paper considers an interesting problem and presents several interesting ideas and therefore believe the paper is worth sharing with the AABI community. However, I agree with reviewers that the experiments are very limited, and I strongly encourage the authors to extend their experiments to make their results more practically relevant. I recommend acceptance.

---

### Decision · Program_Chairs · 2024-05-27

Accept